# Biochemical Responses in *Populus tremula*: Defending against Sucking and Leaf-Chewing Insect Herbivores

**DOI:** 10.3390/plants13091243

**Published:** 2024-04-30

**Authors:** Filip Pastierovič, Alina Kalyniukova, Jaromír Hradecký, Ondřej Dvořák, Jan Vítámvás, Kanakachari Mogilicherla, Ivana Tomášková

**Affiliations:** 1Faculty of Forestry and Wood Sciences, Czech University of Life Sciences Prague, Kamýcká 129, Suchdol, 165 00 Praha, Czech Republic; diuzheva@fld.czu.cz (A.K.); hradecky@fld.czu.cz (J.H.); dvorak18@fld.czu.cz (O.D.); vitamvas@fld.czu.cz (J.V.); or chari.biotech@gmail.com (K.M.); tomaskova@fld.czu.cz (I.T.); 2ICAR-Indian Institute of Rice Research (IIRR), Rajendra Nagar, Hyderabad 500030, Telangana, India

**Keywords:** aphids, carotenoids, chlorophylls *a* and *b*, polyphenolic compounds, proline, spongy moths

## Abstract

The main biochemical traits were estimated in poplar leaves under biotic attack (aphids and spongy moth infestation). Changes in the abundance of bioactive compounds in genetically uniform individuals of European aspen (*Populus tremula*), such as proline, polyphenolic compounds, chlorophylls *a* and *b*, and volatile compounds, were determined between leaves damaged by sucking insects (aphid—*Chaitophorus nassonowi*) and chewing insects (spongy moth—*Lymantria dispar*) compared to uninfected leaves. Among the nine analyzed phenolic compounds, only catechin and procyanidin showed significant differences between the control leaves and leaves affected by spongy moths or aphids. GC-TOF-MS volatile metabolome analysis showed the clear separation of the control versus aphids-infested and moth-infested leaves. In total, the compounds that proved to have the highest explanatory power for aphid-infested leaves were 3-hexenal and 5-methyl-2-furanone, and for moth-infested leaves, trans-α-farnesene and 4-cyanocyclohexane. The aphid-infested leaves contained around half the amount of chlorophylls and twice the amount of proline compared to uninfected leaves, and these results evidenced that aphids influence plant physiology more than chewing insects.

## 1. Introduction

Global climate change has far-reaching effects on all levels of ecosystems, with a predominantly adverse impact on forests around the world. On a global scale, more than 50% of tree damage can be attributed to biotic factors, with insect herbivores emerging as significant stressors in this context [1]. Although the complex interplay between host trees and insect herbivores is difficult to predict, there is a consensus that the negative changes favour insects [2]. Higher temperatures and drought weaken the trees, thus promoting insect abundance and geographical spread, as well as usually shortening their generation time [3]. Nevertheless, plants have the constitutive and induced defence needed to cope with unpredictable changes in the environment. The constitutive defence is based mainly on the substances that form a mechanical barrier against insects (suberin in cell walls or a waxy layer in the epidermis). Induced defences suggest that individual plants have the capacity to alter their chemical phenotype in reaction to biotic stress, thereby potentially exerting a decisive influence on species interactions over ecological and evolutionary timescales, through modifications in interactions [4]. Research has largely focused on exploring these interactions between a plant and a single attacker, representing a pivotal initial stage in unraveling the chemical ecology of plants. However, as our technical capabilities advance, it becomes imperative for us to delve deeper [5].

Photosynthetic pigments are an essential part of the primary phase of photosynthesis, where ATP and NADPH are generated to later fix CO_2_ into carbon products. Therefore, a decrease in chlorophyll *a* and *b* due to various stress factors endangers the survival of the plant by decreasing the assimilation rate and stomatal conductance [6]. On the other hand, the total concentration of carotenoids can increase by up to five times under higher irradiance or in response to biotic attack, as the conversion of violaxanthin to zeaxanthin within the xanthophyll cycle helps to scavenge newly emerging oxygen radicals [7]. Proline, a proteinogenic amino acid with signal function, is on the rise in the case of insect larvae that feed on *Populus* leaves [8]. The higher level of proline enhances the NADP+/NADPH ratio, increasing the oxidative pentose phosphate pathway (OPPP) and producing phenolic compounds directly involved in plant defence [9]. Through underground and aboveground communication channels in the plant, proline serves as a stress marker in the aboveground biomass after root damage caused by, for example, *Melolontha melolontha* [8]. Despite the recorded positive effect of proline on the development of aphid populations, it appears that at higher levels, it acts as a limiting factor [10,11,12]. 

The induced defence is based on the metabolites that emerge several minutes after an insect attack [13]. From the four basic pathways of the secondary metabolism (i.e., phenols, flavonoids, terpenes and nitrogen/sulfur) [14], most of the structural and defence compounds are metabolized via the shikimate–phenylpropanoid pathway. Phenolic glycosides, hydroxycinnamates, flavonoids, and condensed tannins are accumulated in the case of biotic stress [15]. Every plant has its typical profile of secondary metabolites with antifeeding activity or another adverse effect on larval development. In poplar (*Populus*) leaves, phenolic glycosides are very common compounds. Catechin, rutin, and quercetin are responsible for a reduced *Lymantria* larvae weight or prolonged larval development [16]. Although these substances are generally present, their concentration increases after an insect attack [17]. Several studies have also confirmed the induction of phenolic glycosides after an insect attack [15,18,19,20]. The induction of phenolics in trees depends on several factors, including the specific tree species, its genotype, and the species of insect herbivore responsible for the attack [17].

Plants, and especially trees, are the largest source of volatile organic compounds (VOCs) worldwide, both in stressed and non-stressed conditions [19,21]. These compounds are important in plant–insect (including pollinators) or plant–plant communication [22,23,24,25,26,27,28]. Insect herbivory changes the rate of plant VOC emission and the types of compounds emitted [29]. Herbivore-induced plant volatiles (HIPV) are stress-induced VOCs released as a response of the plant metabolism to herbivory [11,17]. The type of feeding damage affects the VOCs produced; leaf chewing generally induces jasmonic acid production, while phloem-sucking insect herbivores tend to induce salicylic acid-mediated signalling pathways. Causality between the induction of VOCs after an attack by an insect herbivore has been demonstrated many times [30,31,32]. 

Biotic stress alters the pigment, proline, and phenolic compounds in Fabaceae species [33], *Ulmus* [34], and *Populus* [35]. Nonetheless, a number of unresolved issues persist with respect to the variability in responses to the different feeding-type behaviours exhibited by insects, as well as the explanatory power of particular primary and secondary metabolites. Based on this, we established an experiment using a genetically uniform line of European aspen (*Populus tremula*). In this study, we attempt to provide material for a thorough understanding of the metabolic manifestations following an attack by an insect herbivore, starting at the level of changes in photosynthetic pigments, progressing to changes in the concentration of the proteinogenic amino acid proline and changes in the induction of secondary metabolism products like phenolic compounds and volatile organic compounds.

## 2. Results 

### 2.1. Photosynthetic Pigments

The control values for chlorophylls *a* and *b* showed the highest value of all treatments. The content of chlorophyll *a* was up to 50% lower in leaves infested by aphids (2.1 ± 0.5 mg.g^−1^ of FW) compared to the control leaves (3.8 ± 0.6 mg.g^−1^ of FW), and about 20% in leaves infested by moths, with the same pattern in chlorophyll *b*. In contrast, carotenoids had the lowest values in the control. The carotenoid content was double in leaves infested by moths and higher by about 40% in aphid-infested leaves (Figure 1). There are significant differences (*p* < 0.01) among the control, aphid and spongy moth treaments for both chlorophylls and carotenoids, with the exception of the control and months-infested treatments for chlorophyll *a* (LSD Fisher’s test). 

### 2.2. Proline

For proline, the values in moth-infested leaves were lower compared to the control leaves (10.0 ± 3.4 and 15.8 ± 5.1 µg.g^−1^, respectively). But for aphid-infested leaves, a proline content was twice as high as that of the control leaves was recorded. No significant differences between the control leaves and moth-infested leaves were found (Figure 2). The difference between the control and aphid-infested leaves was significant. 

### 2.3. Polyphenolic Compounds Allocated in Damaged Poplar Leaves

Compared to a significant decrease in photosynthetic pigments and an increase in carotenoids, the response of phenolic compounds in damaged leaves was not uniform. Procyanidin and catechin showed statistically significant differences for aphid and moth-infested leaves compared to the control leaves. The significantly lowest level of procyanidin was recorded in leaves infested by moths. The procyanidin content was less than half of that of the control leaves. Similarly, the concentration of catechin in moth-infested leaves was lower compared to the control leaves, but not significantly different between the control and aphid-damaged leaves. There was a significant difference in procyanidin between the aphid and spongy moth infestations, where the aphid-infested leaves had the highest concentration (Table 1).

### 2.4. Volatile Compounds Released from Poplar Leaves

The profiles of the volatile compounds emitted from leaves and measured via SPME-GC×GC-TOF-MS were aligned and a data table was created, where the area of the quantification ion (unique mass from deconvoluted mass spectrum of signal) is provided for each of the 304 recorded chromatographical signals. This data table was then evaluated using PCA and OPLS-DA. The initial PCA, explaining 50% of the variance in data, showed a tendency for the separation of samples according to infestation (Figure 3).

OPLS-DA models were created focusing on the difference between the control samples and those infested by spongy moths or aphids. By comparing the control and aphid-infested samples (Figure 4), the model parameters (R^2^X_cum_ = 0.23, R^2^Y_cum_ = 0.97, Q^2^_cum_ = 0.90) show the good separation and predictive power (based on internal cross-validation) of the model. This indicates strong and reproducible damage to poplar leaves by aphids. The ten compounds most responsible for this separation were selected from the VIP plot (Variable Importance Plot, not shown). The VIP values are reported together with their standard error (Table 2). Interestingly, the variability of some compounds (high in control samples) decreased after infestation, showing that infestation with aphids changes the profile of volatiles to a more homogenous one.

The same approach was also used for the spongy moth-infested samples (Figure 5). OPLS-DA in this case also provided good separation power but showed a much lower predictive ability, as can be seen from the model parameters after internal cross-validation (R^2^X_cum_ = 0.15, R^2^Y_cum_ = 0.91, Q^2^_cum_ = 0.49); this probably shows that the damage of foliage was not as extensive as in the case of aphids. The compounds with the most decisive power for the presented separation are listed in Table 3.

## 3. Discussion

After insect attack, the allocation of organic substances changes. The primary metabolism (the main consumer of carbon in plants) ensures the conditions for an efficient secondary metabolism, on which the plant’s defence depends [36]. Hughes [37] hypothesized that plants are constitutionally equipped with a flavonoid metabolism to defend against herbivores, which was later confirmed at the level of mammalian herbivory [38] and more recently at the level of insect herbivory [17]. Rutin formed via the phenylalanine pathway, together with quercetin, coumaric acid, kaempferol, and chlorogenic acid, exhibit a constant concentration independent of herbivore attack [39]. Rutin is a flavone glycoside known to delay insect moulting and cause death [40]. This compound can also prolong the developmental cycle of Lepidoptera and cause higher larval mortality [41,42].

Considering other polyphenolic compounds, chlorogenic acid significantly contributes to constitutive resistance in insects, as found in thrips-resistant chrysanthemums [43]. The polyphenolics produced by the plant and digested by the moth larvae cause an increase in oxygen radicals in the larvae midgut, using the redox balance of glutathione (GSH) towards a higher percentage of its oxidized form (glutathione disulfide; GSSG). The higher ratio of GSSG/total GSH in third-instar moth larvae than in fourth-instar moth larvae suggests a difference in sensitivity to chlorogenic acid or phenolics generally [18]. The enduring interplay between insects and plants, as extensively documented by Agrawal [4], underscores the pivotal role that fundamental genome interactions play in the development of both plant species and herbivores. A common response to insects is an increase in tannin for chewing insects and also for aphids [44]. The more we understand about the plant metabolism under stress, the more apparent it is that the product of the plant metabolism undergoes dynamic coherent processes, of which only a fraction of the many associated individual secondary metabolites and biosynthetic pathways are observable [45]. Nevertheless, the online monitoring of all discussed variables is complicated.

The ability of aspen genotypes to synthesize, accumulate, and store catechin as well as procyanidin (condensed tannins) is genetically determined [46], determining the extent of condensed tannin induction within the population or genome [47]. The results of our research confirm that aphid infestations have many similarities with fungal pathogens [48], whereby sucking phloem sap aphids also induce the salicylic acid–hormonal pathway and thus generate reactive oxygen species [49]. The contribution of flavan-3-ol (whose parts are catechin and procyanidin) to plant defence was proved for microbial pathogens, insects and mammalian herbivores with the direct involvement of salicylic acid [23,50,51]. In mature poplar, spongy moths have only a marginal effect on the accumulation of low-molecular-weight flavan-3-ols, with an increase of 10% in the bark and even a decrease of 10% in the leaves [52]. A decrease in catechin and procyanidin was observed in the case of leaves infested by spongy moths, probably because of an interruption of the central or lateral veins of the leaf. This phenomenon was initially documented during the leaf-feeding activities of various leaf-chewing insect species [53,54,55]. This mechanical damage to the veins interrupts the flow of phloem, toxins, antifeedants, and other secretions. According to the Herms and Mattson [56] theory, in addition to mechanical damage, the allocation of resources for defence in the case of huge and quick damage at the level of a single leaf can also be a persistent dilemma for plants. Phenolic compounds derived via the common phenylpropanoid pathway perform as a signalling molecule and can act as agents in plant shielding [57]. The result of taxifolin induction in our study shows the same dynamics in the treatment of aphids as in the attack by fungal pathogens, which is observed in the works of Ullah et al. [51] and Hammerbacher et al. [58]. According to the metabolic pathway, taxifolin originates from the metabolism of the phenylalanine pathway and is a precursor of catechin, proanthocyanidins and quercetin [51]. The results of the decrease in the concentration of these substances in the treatment of aphids show the activation of this defense mechanism in *Populus tremula*. In contrast, a decrease in the concentration of these metabolically related compounds was observed in the treatment of moths. The treatment of moths according to the increased concentration of ferulic acid shows the activation of hydroxycinnamic acid amides (HCAAs). This group of coumpounds is widely distributed in plant secondary metabolites and is often referred to as one of the major phenylpropanoid metabolites [59]. HCAAs, including ferulic acid, have been discovered to exert inhibitory effects on acetylcholinesterase, an enzyme crucial in the molting process of the rice weevil (*Sitophilus oryzae* L.) [60]. Despite the general consensus, the combined mixture of allelochemicals significantly improves plant defense against insect herbivores. Nevertheless, there are populations of *Lymantria dispar* that show excessive tolerance to the tannins contained in plant tissues [61].

Sesquiterpene (E,E)-α-farnesene was one of the most important volatile compounds for both types of infestation in our research, and was also previously found in leaves after spongy moth damage [15]. Other investigations also indicate that this substance is a fairly typical VOC produced in response to insect herbivory [62,63,64]. However, the biological interpretation of the effect is still unclear. James and Grasswitz [65] gives examples of parasitic wasps (*Anagrus* spp.) attracted by farnesene. Furthermore, studies by Beale et al. [66] mention the possibility of the influence of farnesene on reducing the occurrence of viruses transmitted by aphids. This compound was the most important for differentiation between the control and spongy moth-infested leaves, while the fifth most important for aphid-infested poplars versus the control ones. A similar behaviour was observed for dendrolasin, a compound derived from farnesene via biosynthesis [67]. In our study, the presence of green leaf volatiles (GLVs) like 3-hexanal indicates its metabolic availability in the poplar genome. Due to the high sensitivity of the olfactory system of insect herbivores [68], they may act as active attractants to the natural enemies of insect herbivores [69]. Interestingly, the higher emission of 3-hexanal observed during the night hours corresponds to the nocturnal activity of *Lymantria dispar* caterpillars [70]. There is not much information about the biological mechanisms and the effect on insect herbivory. Dendrolasin, as a component of essential oils, performs antimicrobial activity in plants [71,72].

Moreover, the production of primary metabolites is also compromised, as at least some chlorophyll must be renewed daily. Chlorophyll *a* and *b,* in response to phytophagous insects, is decreasing in many different plant species [33,73]. For example, the decrease in chlorophyll *a* and *b* observed in *Citrus* leaves correlates positively with the density of colonization of *Coccus hesperidus* [74]. As we observed, the chlorophyll decrease depends on the insect species. The negative impact of aphids was greater on the content of chlorophylls compared to spongy moths, with a 50% decrease recorded compared to a 15% chlorophyll loss in moths. The chlorophyll content in plant tissues is a key factor in interactions between plants and insects [74]. Changes in chlorophyll concentrations occur in response to a wide range of stresses, including biotic stresses such as insect feeding and pathogenic infections [33]. Our results support the observations of Huang et al. [75], as we confirmed a relative decrease in the chlorophyll content in *Populus tremula* in response to both sucking and chewing insect attacks. Photosynthesis is the main source of reactive oxygen species (ROS) in light, and reactions in chloroplasts produce a variety of ROS forms, including singlet oxygen, superoxide, and hydrogen peroxide, at high rates even under optimal conditions [76,77]. ROS have a detrimental effect on the plant DNA and protein complexes. At the same time, insect herbivory has a negative impact on the leaf water status, resulting in stomatal closure [78]. Therefore, it is possible that plants suffering from insect herbivory may also experience excess excitation energy caused by an excessive concentration of singlet oxygen [79]. Given the extensive crosstalk between light, ROS, and hormonal signalling, this phenomenon is likely to have a strong impact on plant responses to insect herbivores. Considering this, increasing concentrations of certain phenolic compounds, known for their scavenging activity in eliminating ROS, appear to be advantageous [80]. Next to both chlorophylls, carotenoids are also involved in the trapping of light in the first phase of photosynthesis. However, carotenoids participate in other physiological functions, e.g., antioxidative activity is usually enhanced after an insect attack [81]. The response of carotenoids is not as straightforward as in the case of chlorophyll, depending on tree species and insect density [74]. The level of carotenoids follows an irregular curve, with an increase after infestation and a decrease over a prolonged time [74], or simply a decrease [73]. In our study, the carotenoid level was enhanced in both aphid- and moth-attacked leaves, and their level on moth-attacked leaves was double compared to the control. The increase in carotenoids was 40% higher in aphid-attacked leaves compared to the control leaves. Other studies have shown that the proline content and peroxidase activity reached their peak after 7 days of exposure to sucking insects from the *Pseudococcus* family [82]. Peroxidase scavenges oxygen radicals, as do carotenoids [83]. Consistent with the published results, proline increased after insect attack: the proline level increased by two times in aphid-attacked leaves, while the increase in proline in moth-attacked leaves was lower (by 40%). The work of Lackner et al. [8] provides insight into the dynamics of proline in *Lymantria dispar*, which, according to their observations, has a phagostimulatory effect; this implies that leaves with higher levels of proline are preferred by spongy moths. According to our results, the activity of this herbivore does not induce increased levels of proline in *Populus tremula*. Proline is considered a reliable marker indicating plant stress due to drought [84,85], reaching up to three times the concentration [86]. In this context, there is a potentially greater risk of spongy moth attack in poplar stands suffering from drought. As part of the aphid treatment, there was apparently a significant increase in the proline content due to the consumption of phloem sap by aphids. 

## 4. Materials and Methods

### 4.1. Plant Material

European aspen (*Populus tremula*) seeds were used as the initial plant material. Genetically homogeneous individuals were employed to achieve statistically significant results. These individuals were propagated utilizing somatic embryogenesis techniques. The seeds were obtained by the controlled crossing of parent trees (locations: Czech Republic, Sušice (Svatobor), and Ore Mts. (Fláje), 40–50 years old), which was carried out in early spring 2019. From five to seven days after seed collection, the seed material was used for in vitro propagation. In total, 218 seedlings sprouted, from which individual number 22 was selected because it propagated best in in vitro culture. Seeds of *P. tremula* were washed in 200 mL of distilled water with the addition of 1–2 drops of Tween 20^®^ (Sigma-Aldrich) for 10–15 min. The seeds were then sterilized in 0.1% HgCl_2_ for 6 min, rinsed three times in sterile distilled water, and placed in 230 mL jars containing 30 mL of Murashige and Skoog (MS) medium [87] solidified with 8 g.L^1^ Danish^®^ agar (Carl Roth), containing 100 mg·L^−1^ myo-inositol, and supplemented with 1 mg·L^−1^ 6-benzylaminopurine (BAP). After the adjustment of pH to 5.7, the medium was sterilized in an autoclave at 121 °C and 118 kPa for 30 min. The explants were cultivated under a 16/8 h light/dark photoperiod (photosynthetic photon flux density 35 ± 2 μmol. M^−2^·s^−1^ cool white fluorescent light), at a temperature of 22 ± 1/20 ± 1 °C (light/dark). The first seeds began to germinate after 1 week on the medium. Most germinated in 2–3 weeks after deployment on the MS medium. Over the next 2–3 months, the plants formed new longer shoots, which were further used for multiplication. The germinated seeds and newly sprouted shoots were regularly subcultured every 2–3 weeks on the same medium until sufficient plant material was obtained for rooting. Dry, brown, and contaminated explants were discarded during the in vitro cultivation.

In vitro rooting was performed on segments about 1.5–2.5 cm long with at least three buds. The shoots developed in vitro were rooted on half-strength MS medium (Murashige and Skoog, 1962) with the addition of 0.5 mg·L^−1^ indole-3-butyric acid (IBA). The first roots began to develop after about 4 weeks on rooting medium and, after 6–8 weeks of cultivation, the rooted shoots were used for ex vitro transfer.

Well-rooted shoots were removed from the cultivation jars and the roots were washed with water to remove residues from the culture medium. The plants were transferred into a sterile substrate (peat and perlite, Forestina, Czech Republic) within plastic pots (7 × 7 × 8 cm), watered, and then treated with 1% Previcur Energy^®^, Bayer Garden, Germany). The plants were cultivated in an air-conditioned room under a photoperiod of 16/8 h (day/night, photosynthetic photon flux density of 35 ± 2 μmol·m^−2^·s^−1^ cool white fluorescent light) and a temperature of 22 ± 1/20 ± 1 °C (day/night). They were acclimated by gradually decreasing the air humidity from 95% to 60%. One month after ex vitro transfer, the plants were transferred to cultivation boxes. During ex vitro cultivation, the plants were fertilized using NPK fertilizer once every 2 weeks during vegetation growth.

### 4.2. Insect Breeding

Eggs of spongy moth (*Lymantria dispar*) were obtained from sterile laboratory cultures from the University of Natural Resources and Life Sciences, Vienna. After hatching, the larvae were fed in sterile petri dishes with nutritionally balanced agar. In the experiment, caterpillars were used after the fourth molting. To enhance the feeding activity, they were incubated in darkness and deprived of food for 48 h, considering their nocturnal behavior. The stockbreeding of aphids (*Chaitophorus nassonowi*) was established by catching adult individuals in the wild during the colonization of aspen. Then, the aphids were cultured in sterilized plastic containers for several months. In vitro cultures of poplar individuals were used as food for the aphids. This method of breeding minimizes the variance in possible phytopathological contamination (especially the development of moulds and fungi).

### 4.3. Experimental Design

The strategic goal of this experimental design was to arrange a process that minimized any other effects on aspen individuals. Therefore, through the action of model insect species (leaf-sucking and leaf-chewing insect species), the influence of insect herbivory was the only factor that affected the plant metabolism. The experiments were established in Step-In FytoScope FS-SI growth chambers (Photon Systems Instruments, Drasov, Czech Republic). In the growth chambers, the conditions set simulated the optimal environmental conditions (humidity: 75%; average intensity of Photosynthetic active radiation: 250 µmol.m^−2^·s^−1^; CO_2_ concentration: 415 ppm; day and night period: 2 h of dawn, 10 h of light, 2 h of twilight, 10 h of darkness). The basic structure of the experiment (Figure 6) was made up of three treatments of aspen, where each group had 20 individuals. To prevent intraspecific chemical communication, each group was grown in a separate growth chamber. Group markings: (C) Control—individuals without damage; (A) individuals attacked by aphids (leaf-sucking); (M) individuals attacked by spongy moths (leaf-chewing).

### 4.4. Sample Collection

Sample collection took place five months after the transfer of poplars to ex vitro. Leaf collection took place during the treatment of the damage caused by the 5 spongy moths at the moment at which about 30% of the leaf was eaten within 30 min. The samples in the aphid treatment were harvested after about 4–5 days of colonization (the number of aphids was around 100 in different stages of development) when signs of damage were visible on the leaf. Aphids were removed from the leaf with prepared individual brushes with natural fibres (washed repeatedly in chloroform and stored in sterile aluminium foil). The leaves were cut off and immediately put into liquid nitrogen. In order to minimize the influence of circadian cycles on the plant metabolism, samples were always collected within a time interval of 6 h after dark, and the collection was stopped 2 h before dark in the growth chambers.

### 4.5. Spectrophotometric Measurement

For photosynthetic pigment estimation, the common method of extraction and analysis was used [88], but several steps were modified. First, 10 mg of deep-frozen plant tissue was placed into a 2 mL test tube and homogenized using a mill crusher (Retsch mill, Haan, Germany) for 3 min and 30 oscillations per second. Then, 1.5 mL of acetone and several crystals of magnesium carbonate (to stop the pheophytin formation) were added. Before centrifugation, the pigments were left to elute from the tissue for a few minutes. Then, the samples were centrifuged for 3 min at 13,000 rpm. The supernatant was collected in the new test tube and supplied with 5 mL of acetone prior to spectrophotometric analysis. The absorbance was measured at 663 nm, 646 nm, and 470 nm for the estimation of chlorophylls *a*/*b* and the total carotenoid content using a spectrophotometer DR6000 (HACH Company, Loveland, CO, USA).

The following equations were used for calculation:$$c_{chl\;a}=12.25\;A_{662}-2.81\;A_{647}$$
$$c_{chl\;b}=20.13\;A_{647}-5.03\;A_{664}$$
$$c_{total\;carotenoids}=(1000\;A_{470}-3.27\;c_{chl\;a}-104\;c_{chl\;b})/198$$

The proline content was determined according to Bates [89] with the following modifications. For the analysis, 20 mg of the plant tissue was weighed (fresh weight FW) and homogenized in the mill crusher for 3 min. Then, 300 μL of 3% sulfosalicylic acid was added and mixed well using a vortex for 30 s. Next, the samples were centrifugated for 5 min at 13,000 rpm. Then, 160 µL of supernatant was transferred to the new test tube. After this, 160 µL of glacial acetic acid and 160 µL of ninhydrin solution were added (ninhydrin solution was prepared from 0.25 g of ninhydrin, 6 mL of glacial acetic acid, and 4 mL of phosphoric acid). The solution was vortexed and inserted into a heat block (Major Science, Taoyuan City, Taiwan) at 95 °C for one hour. Then, the samples were left to cool down. Once cool, 320 μL of toluene was added and vortexed for 30 s for the extraction of proline to the organic phase. After the phase separation, the absorbance of the upper phase was measured at 520 nm against toluene.

### 4.6. Extraction Procedure for Determination of Polyphenolic Compounds

Each sample consisted of 20 mg of lyophilized and homogenized plant tissue, meticulously weighed. Subsequently, 500 μL of a methanol/water solution (70:30 *v*/*v*, sourced from Honeywell, Offenbach, Germany) was added to each sample. Then, the sample was mixed well using a vortex and placed into an ultrasonic bath with ice for 10 min. After, the samples were centrifuged at 13,000 rpm for 10 min at 4 °C. The supernatant was collected and filtered through a PTFE syringe filter (0.22 µm) prior to LC-MS-qTOF analysis.

### 4.7. LC-MS-qTOF Analysis of Polyphenolic Compounds

LC-MS-qTOF analysis was carried out using an Agilent 1290 Infinity II coupled with an Agilent 6546 LC/MS QTOF system (Agilent, Santa Clara, CA, USA) and Zorbax Eclipse Plus C18 column (2.1 × 50 mm, 1.8 µm), (Agilent, USA). Mobile phase A contained 0.05% formic acid and mobile phase B consisted of acetonitrile. The gradient elution was as follows: 0–1 min, 100% A; 1–7 min, 35% A; 7–8 min, 100% A; 8–8.01, 100% A; and 8.01–10 min, 100% A. The flow rate of the mobile phase was set at 1.1 mL minute^−1^ [35]. The column temperature was set at 35 °C. The injection volume was 1 µL. The system was operated in negative ionization mode. The optimization of qTOF parameters was previously performed using the standards of polyphenolic compounds. The qTOF parameters were as follows: scan range, 100–1000 *m*/*z*; drying gas temperature, 350 °C; sheath gas flow rate, 12.0 L/minute; sheath gas temperature, 400 °C; capillary voltage, 5.0 kV; nozzle voltage 0.9 kV; fragmentor, 140 V; and collision energy at 10, 20 and 40 eV. MS/MS data were acquired at a scan range of 50–800 *m*/*z* using a 0.5 min retention time window, an isolation window of 1.3 amu, and an acquisition rate of 2 spectra s^−1^. During the analysis, two reference masses (112.9855 *m*/*z* and 966.0007 *m*/*z*) were continuously measured for mass correction. Data collection was carried out using Agilent Mass Hunter Acquisition software Workstation Plus 11.0. Data analysis was performed using Qualitative Analysis 10.0 and Q-TOF Quantitative analysis.

### 4.8. GC×GC-TOF-MS Metabolomic Analysis

The fingerprints of poplar leaf volatiles were collected using solid-phase microextraction (SPME) coupled to two-dimensional comprehensive gas chromatography and time of flight mass spectrometry (GC×GC-TOF-MS). From each freeze-dried sample, 200 mg was placed into a 10 mL headspace glass vial and sealed with a screw lid with PTFE septum. Volatiles from the sample headspace were collected after 10 min of incubation at 50 °C using an SPME fiber with an divinylbenzene/carboxen/polydimethylsiloxane coating from Supelco (Bellefonte, PA, USA).

An agilent gas chromatograph 7890B (Agilent Technologies, USA) was equipped with a HP-5MS UI capillary column (30 m, 0.25 mm i.d., 0.25 μm film thickness) for one-dimensional separation and coupled to BPX-50 (SGE, Victoria, Australia, 1.5 m, 0.1 mm i.d., 0.1 μm film thickness) for two-dimensional separation using a consumable free modulator. Helium at a flow of 1 mL min^−1^ was used as a carrier gas. The spitless injection was applied to a hot (265 °C) split/splitless injector. After 2 min of solvent delay, the oven temperature was increased from an initial 40 °C at a rate of 10 °C min^−1^ to 280 °C. The second dimension GC oven and modulator followed this temperature program, with 5 °C and 15 °C offset, respectively. With a hold time of 1 min, the GC run took 27 min.

In initial data processing, the peak find and automated spectral deconvolution algorithms were performed in ChromaTOF SW (LECO, St. Joseph, MI, USA). Then, the peak alignment tool (Statistical Compare, LECO) was used to align signals with S/N higher than 150, with a maximal retention time deviation between samples of no more than 5 s and consequently a spectral similarity between signals in samples higher than 75%.

For the tentative identification of compounds, a spectral comparison of the measured and deconvoluted spectrum with mass spectra in the National Institute of Standards and Technology mass spectral library (NIST 2017) library was performed. To support this identification, retention indexes from NIST were used.

### 4.9. Statistical Evaluation

A statistical evaluation of the GC pre-cleaned and centered log-ratio (CLR) transformed metabolomic data was performed using Simca 17.0 SW (Sartorius Stedim Data Analytics AB, Umeå, Sweden), where Principal Component Analysis (PCA) and Orthogonal Partial Least Square Discriminant Analysis (OPLS-DA) were used. PCA was used mainly to judge the obtained data concerning the measurement quality and to remove outlying samples. For marker selection, OPLS-DA was used. In this model, samples were classified based on the species of insect. 

One-way ANOVA was used for the estimation of statistically significant differences among the phenolic compounds, proline and photosynthetic pigments. Data normality and variance homogeneity were tested before the analysis. To reveal the differences between the control and treatments with aphids and moths, a post hoc Scheffe test was applied. Statistica 14.1.0 (Tibco Software Inc., Palo Alto, CA, USA) was used to test the differences at the level of 0.05.

## 5. Conclusions

In this investigation, our focus was on providing a comprehensive understanding of plant responses to insect herbivory, spanning from the level of photosynthetic active pigments (the primary energy contributors to metabolism) to alterations in the proteinogenic amino acid proline, and extending to the most metabolically intricate compounds in the context of herbivory by both sucking and leaf-chewing species (graphical abstract). Even in the case of an in vitro experiment designed with genetically uniform plants for an exact description, the complexity of the connections, flow, and interaction of substances and influences is evident. Currently, a comprehensive systematic framework is lacking. However, with the impact of genetic interactions, chemotypes, and metabolic strategies, there are ample opportunities to enhance our understanding of the dynamics in plant host–herbivore interactions. Despite the abundance of publications and studies, there are many opportunities to expand our knowledge of plant host–herbivore interaction dynamics. In this study, we observed lower concentrations of chlorophyll a and b in *Populus* leaves attacked by moths and aphids. At the same time, the leaves damaged by moths and aphids had twice the concentration of carotenoids compared to the control. In addition, it was found that in leaves attacked by aphids, an increased concentration of proline was observed compared to the control. We also observed the same concentrations of flavan-3-ol compounds (like catechin and procyanidin), whereas trans-α-farnesene and 3-hexenal were the most differentiated compounds in moth and aphid-infested leaves compared to the control leaves, respectively.

## Figures and Tables

**Figure 1 plants-13-01243-f001:**
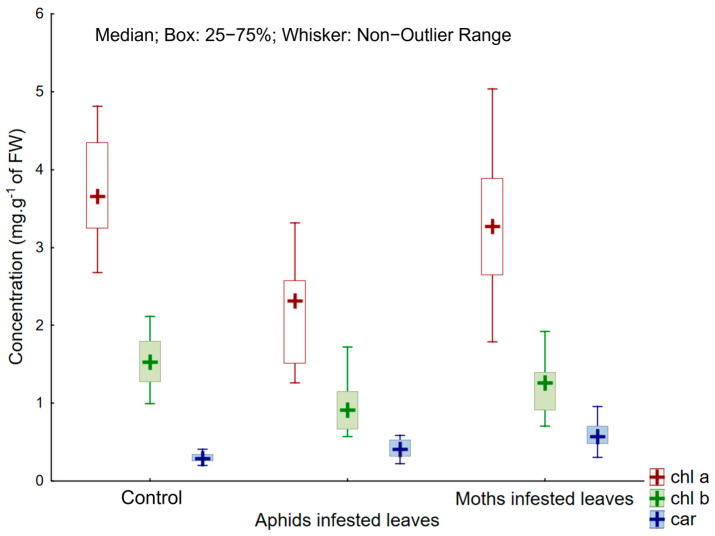
Concentration of chlorophyll *a* (chl *a*), chlorophyll *b* (chl *b*) and carotenoids (car) in aphid-infested leaves (A) and moth-infested leaves (M) compared to the control leaves (C).

**Figure 2 plants-13-01243-f002:**
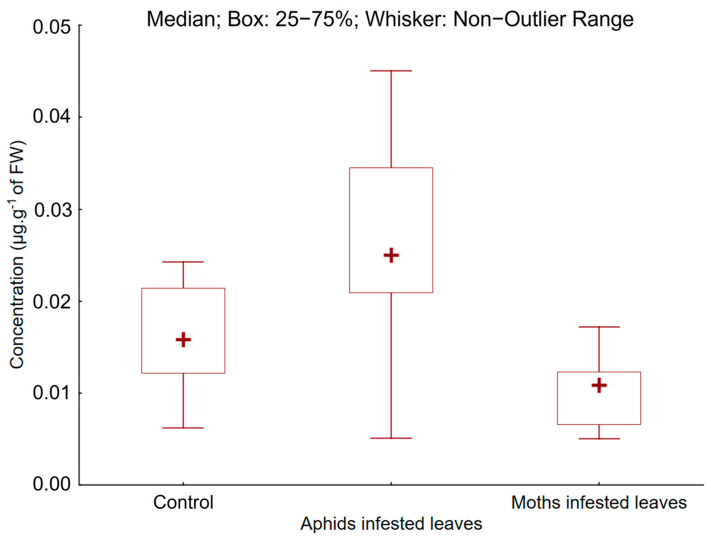
Concentration of proline in aphid-infested leaves (A) and moth-infested leaves (M) compared to the control leaves (C).

**Figure 3 plants-13-01243-f003:**
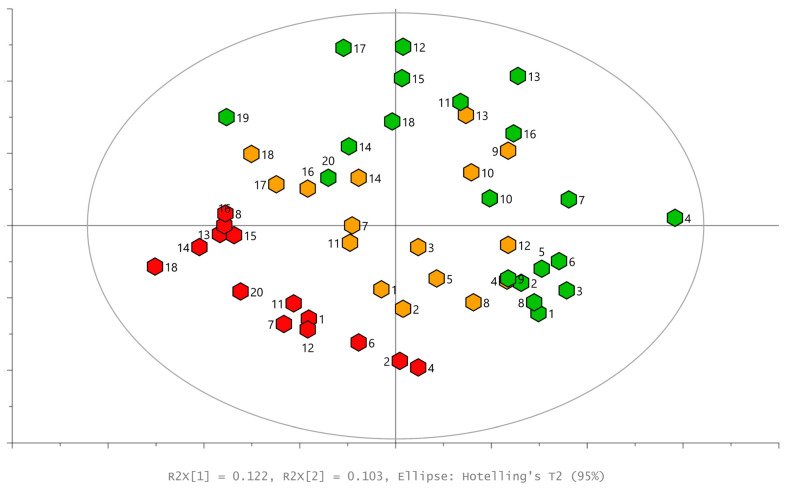
PCA scores plot, visualized according to the first two principal components; the numbers of samples are provided close to each hexagon, presenting an individual sample: green—control leaves, red—aphid-infested leaves, orange—moth-infested leaves. Hotteling’s T^2^ ellipse α = 0.05.

**Figure 4 plants-13-01243-f004:**
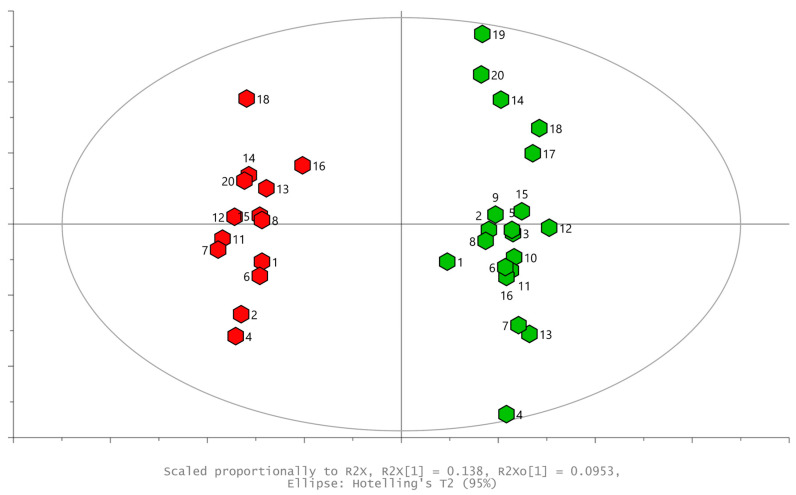
OPLS-DA scores plot showing separation of control leaves (green) and aphid-infested (red) volatiles in poplar leaves; model parameters (R^2^X_cum_ = 0.23, R^2^Y_cum_ = 0.97, Q^2^_cum_ = 0.90); Hotteling’s T^2^ ellipse α = 0.05.

**Figure 5 plants-13-01243-f005:**
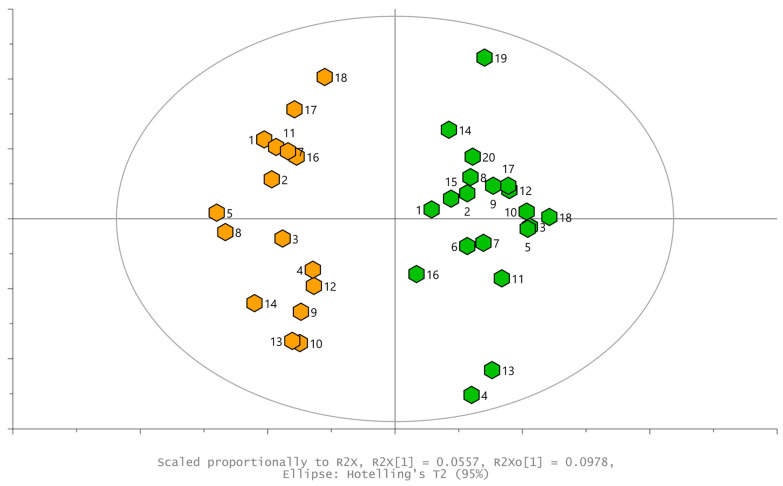
OPLS-DA scores plot showing separation of control (green) and moth-infested (yellow) poplar leaves; (R^2^X_cum_ = 0.15, R^2^Y_cum_ = 0.91, Q^2^_cum_ = 0.49); Hotteling’s T^2^ ellipse α = 0.05.

**Figure 6 plants-13-01243-f006:**
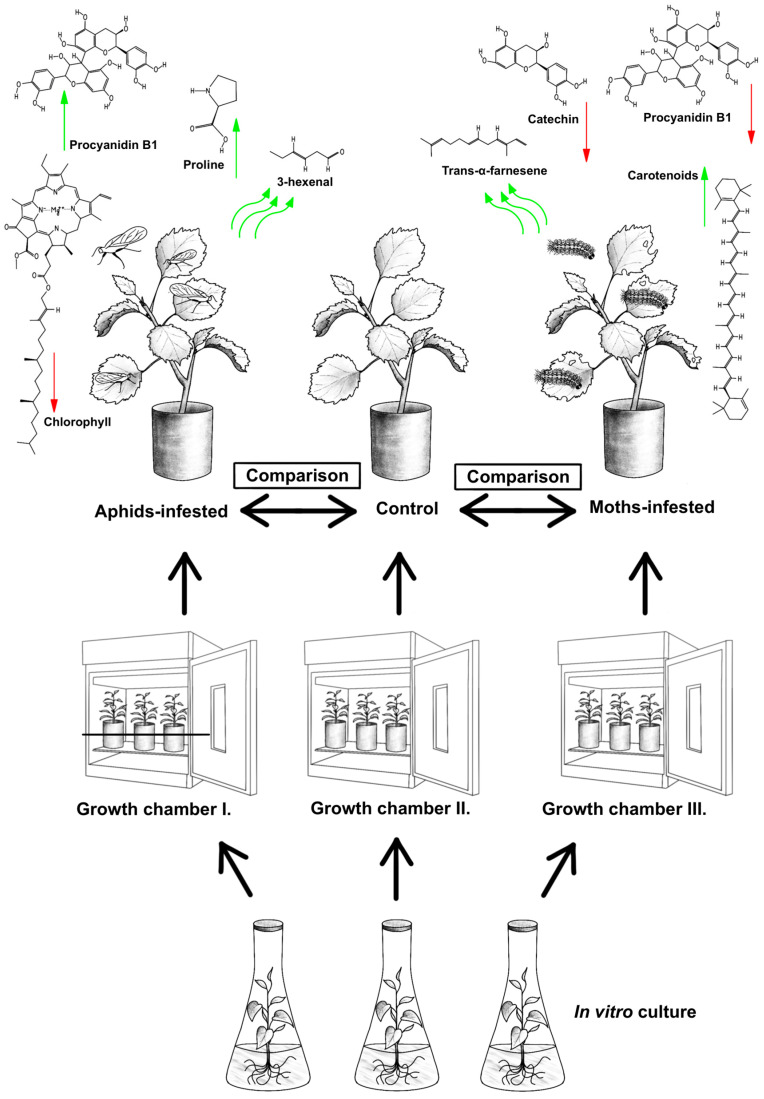
Graphical representation of the experimental design. Green arrows show individual compounds increase, red arrows indicate decrease.

**Table 1 plants-13-01243-t001:** Overview of the estimated concentrations of the investigated phenolic compounds in leaves in μg.g ^−1^ of dry weight (DW). N = 20. Asterisks denote a statistically significant difference.

Compound	Mean ± Standard Error (μg g^−1^ of DW)		
	Control Poplar Leaves	Moth-Infested Leaves	Aphid-Infested Leaves
4-coumaric acid	21.4 ± 2.4	20.1 ± 3.7	30.5 ± 5.7
Rutin	29.6 ± 1.6	31.6 ± 2.1	29.1 ± 1.4
Catechin	37.6 ± 4.2	9.6 ± 1.5 *	41.0 ± 6.6 *
Taxifolin	2.6 ± 0.7	0.32 ± 0.03	1.5 ± 0.3
Procyanidin B1	23.4 ± 2.6	10.5 ± 1.3 *	45.6 ± 10.3 *
Chlorogenic acid	1231.6 ± 62.2	1569.7 ± 83.7	1231.0 ± 84.5
Ferulic acid	20.1 ± 3.6	25.1 ± 3.3	14.3 ± 1.9
Kaempferol	0.7 ± 0.1	0.6 ± 0.1	0.8 ± 0.1
Quercetin	2.4 ± 0.5	2.3 ± 0.3	2.2 ± 0.4

**Table 2 plants-13-01243-t002:** The most decisive volatile compounds for the control leaves and aphid-infested leaves. The “*” symbol represents multiplication.

Compound	VIP	VIP cvSE * 2.44693	Spectral Similarity (%)	RI (calc)	RI (NIST)
3-Hexenal	2.80	0.82	93	800	800
5-Ethyl-2(5H)-furanone	2.74	1.01	81	962	963
Unknown (RI 1358)	2.59	0.46	-	1358	-
2-Hexenal	2.33	0.69	89	848	847
trans-α-Farnesene	2.31	1.47	84	1514	1511
Dendrasaline	2.30	1.06	78	1586	1579
trans-2,4-Hexadienal	2.30	0.83	92	919	913
Hexyl acetate	2.26	0.45	71	1010	1013
Unknown (RI 962)	2.14	0.81	-	962	-
Ethyl 2-oxopropionate	2.11	0.84	71	770	774

**Table 3 plants-13-01243-t003:** Most decisive volatile compounds for the control leaves and moth-infested leaves. The “*” symbol represents multiplication.

Compound	VIP	VIP cvSE * 2.44693	Spectral Similarity (%)	RI (calc)	RI (NIST)
trans-α-Farnesene	2.76	1.38	84	1514	1511
4-Cyanocyclohexene	2.60	0.82	78	1024	1027
Indole	2.50	1.83	80	1306	1300
2-Hexenyl acetate	2.43	1.45	90	1014	1017
Dendrasaline	2.43	1.00	78	1586	1579
Hexyl acetate	2.32	1.35	71	1010	1013
Dihydromyrcenol	2.17	1.19	74	1079	1072
Germacrene D	1.99	1.63	83	1500	1489
3-Hexen-1-ol	1.85	1.22	95	852	856
2-Pentanone	1.75	1.82	88	686	689

## Data Availability

Data are contained within the article and Appendix A.

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
