# Peer review of "Biochemical Responses in Populus tremula: Defending against Sucking and Leaf-Chewing Insect Herbivores"

_plants, 2024, doi:10.3390/plants13091243_

Round 1
Reviewer 1 Report
Comments and Suggestions for Authors
The paper aims to decipher the metabolic response of aspen trees to the attack of a sapsucking (aphid) and a chewing insects (moth) analysing the level change of photosynthetic pigments, proline, polyphenols and VOC’s . The simultaneous analysis of these two insects is a major strong point since the metabolic response of plants to insects with different feeding styles is often quite different and comparison under the same environmental conditions may help to unravel a complex problem.
The general experiment is well designed to answer the aims of the research.
Data in general are clearly presented and the conclusion well supported.
There are only a few oversights and some not very appropriate terms along the text, which are reported in the corrected file.
More specifically :
Line 17: the sentence doesn’t reflect correctly the results
check the measure unit at line 95 (not µgram but mg) and 104
Line 107 the result is to be inverted (Moth less than reference)
The secund column of tables 2 and 3 need to be corrected
Line 320: the experiment is not 3-factorial
Please check the note at 146-8
References are presented in a non-homogeneous way, they should be formatted according to the journal request, but please use italics for all scientific names
Several other minor points are highlighted in the revised file

English is fine,
I suggested some amendements, but the following lines must be rewritten to improve clarity:
ines 53-54; 168-9; 214-6; 258-9; 468-471
Author Response
Reviewer 1
The paper aims to decipher the metabolic response of aspen trees to the attack of a sapsucking (aphid) and a chewing insects (moth) analysing the level change of photosynthetic pigments, proline, polyphenols and VOC’s . The simultaneous analysis of these two insects is a major strong point since the metabolic response of plants to insects with different feeding styles is often quite different and comparison under the same environmental conditions may help to unravel a complex problem.
The general experiment is well designed to answer the aims of the research.
Data in general are clearly presented and the conclusion well supported.
There are only a few oversights and some not very appropriate terms along the text, which are reported in the corrected file.
More specifically:
Answer: Thank you very much, Reviewer 1, for your insights and detailed analysis of our work. All the suggestions from the attached file have been incorporated, and the reference list has been completely revised according to the guidelines for authors. The changes are marked in the docx file with tracked changes and comments.
Line 17: the sentence doesn’t reflect correctly the results
Answer: The sentence has been rewritten for better clarity: “In total, compounds proved to have the highest explanatory power for aphid-infested leaves: 3-hexenal, 5-methyl-2-furanone and for moth infested leaves trans-α-farnesene and 4-cyanocyclohexane.”
check the measure unit at line 95 (not µgram but mg) and 104
Answer: It has been modified.
Line 107 the result is to be inverted (Moth less than reference)
Answer: It has been modified.
The secund column of tables 2 and 3 need to be corrected
Answer: For better clarity and aesthetics of the table, the multiplication sign (*) has been moved to the coefficient line. VIP cvSE * 2.44693 means standard error of VIP value, coming from cross validation of model and mutliplied by factor of 2.44693.
Line 320: the experiment is not 3-factorial
Answer: It has been modified.
Please check the note at 146-8
Answer: We observed this phenomenon when checking abundance of compounds in groups via Variable line plot, but could be done also via comparison of average value and SD for one compound in two groups of samples.
References are presented in a non-homogeneous way, they should be formatted according to the journal request, but please use italics for all scientific names
Answer: The reference list has been completely revised according to the guidelines for authors.
Several other minor points are highlighted in the revised file
Answer: All the suggestions from the attached file have been incorporated
Please see the attachment.

Reviewer 2 Report
Comments and Suggestions for Authors
Manuscript titled “Biochemical Responses in Populus tremula: Defending Against Sucking and Leaf-Chewing Insect Herbivores” studied some of the chemical compound that were known to be involved in plant defense against insect. Here the authors compared these chemicals in Populus tremula plants , when they were treated with chewing and sucking insect under controlled conditions.
The following comment may improve the quality of the manuscripts if authors consider revising the present manuscript based on the suggestions provided in the comments.
Result :
Fig1:
The graph y-axis has wide range of scaling, so it is not clear to see the difference among the condition. To make visible difference, authors may consider to make separate graphs of Chl a, b, car and proline and indicate significant difference with letters or asterisks.
Line 108-109: check the results of the proline content. It is misleading, authors may consider to re-write as, “there was no significant difference between refence and aphid infested proline content, however two fold increase was observed in case of aphid infested samples”.
2.3 Polyphenolic compounds allocated in damaged poplar leaves
Why the authors discussed only Procyanidin, when there are other compounds showing significant difference. For example, Taxifolin in moth infested leaves, and Ferulic acid in aphid infested leaves as compared to the reference leaves.
Re-write the result section including all the significant differences among the conditions.
Table 1: Represent the values with significant difference.
Table S1 is cited in the manuscript.
Discussion
Line 236: Pathogen and insect
The authors are supported their results with previous studies but did not speculate the reasons for these changes and their role in plant defense. For example, insect feeding in this study observed that decrease in chlorophyll content and increase in proline content. Authors should discusses the importance of these changes in plant defense with strong evidences from previous studies.
Authors claimed that the some volatile contents decreased during the aphid infestation. This phenomenon has to be discussed in the context of plant defense against insects.
Materials and methods.
Did authors confirmed the “genetically homogenous” condition of the plants ?
Please write the number and stage of the insects released on to the plants.
Why authors specifically used “Scheffe test” post-hoc test, as their sample number/size of the experiment is equal?
Comments on the Quality of English LanguageThe manuscript requires English language editing to improve its quality. In many places the phrases are difficult to understand, they could be re written simple way for better understanding.
Author Response
Reviewer 2
Thank you very much, Reviewer 2, for the recommendations and comments to improve the quality of the manuscript, which we have incorporated into our study.
Comments and Suggestions for Authors
Manuscript titled “Biochemical Responses in Populus tremula: Defending Against Sucking and Leaf-Chewing Insect Herbivores” studied some of the chemical compound that were known to be involved in plant defense against insect. Here the authors compared these chemicals in Populus tremula plants, when they were treated with chewing and sucking insect under controlled conditions.
The following comment may improve the quality of the manuscripts if authors consider revising the present manuscript based on the suggestions provided in the comments.
Result :
Fig1:
The graph y-axis has wide range of scaling, so it is not clear to see the difference among the condition. To make visible difference, authors may consider to make separate graphs of Chl a, b, car and proline and indicate significant difference with letters or asterisks.
Answer: The mentioned figures were divided according to the proposal for better clarity, the units and axis range were checked.
Line 108-109: check the results of the proline content. It is misleading, authors may consider to re-write as, “there was no significant difference between refence and aphid infested proline content, however two fold increase was observed in case of aphid infested samples”.
Answer: The sentence on this line has been overwritten - due to the higher/lower confusion.
“For proline, the values in moths-infested leaves were lower compared to the reference leaves (10.0±3.4 and 15.8±5.1 µg.g-1, respectively). But for aphids-infested leaves, a twice higher proline content compared to the reference leaves was recorded. No significant differences between reference leaves and moth infested leaves were found (Figure 1). The difference between reference and aphids-infested leaves were significant.“
2.3 Polyphenolic compounds allocated in damaged poplar leaves
Why the authors discussed only Procyanidin, when there are other compounds showing significant difference. For example, Taxifolin in moth infested leaves, and Ferulic acid in aphid infested leaves as compared to the reference leaves.
Answer: According to the proposal, the discussion part was expanded to include taxifolin and ferulic acid.
„The result of taxifolin induction in our study shows the same dynamics in the treatment of aphids as in the attack by fungal pathogens, which is observed in the works Ullah et al. [81] and Hammerbacher et al. [35]. According to the metabolic pathway, taxifolin originates from the metabolism of the phenylalanine pathway and is a precursor of catechin and proanthocyanidins and quercetin [81]. The results of the decrease in the concentration of these substances in the treatment of aphids show the activation of this defense mechanism in Populus tremula. In contrast, a decrease in the concentration of these metabolically related compounds was observed in the treatment of moths. The treatment of moths according to the increased concentration of ferulic acid shows the activation of hydroxycinnamic acid amides (HCAA´s). This group of coumpounds is widely distributed in plant secondary metabolites and is often referred to as one of the major phenylpropanoid metabolites [37]. HCAA´sincluding ferulic acid, have been discovered to possess inhibitory effects on acetylcholinesterase, an enzyme crucial in the molting process of the rice weevil (Sitophilus oryzae L.) [54]. Despite the general consensus, the combined mixture of allelochemicals significantly improves plant defense against insect herbivores. Nevertheless, there are populations of Lymantria dispar that show excessive tolerance to tannins contained in plant tissues [9].”
Re-write the result section including all the significant differences among the conditions.
Answer: Significant results were marked in the "Results" section
Table 1: Represent the values with significant difference.
Table S1 is cited in the manuscript.
Discussion
Line 236: Pathogen and insect
The authors are supported their results with previous studies but did not speculate the reasons for these changes and their role in plant defense. For example, insect feeding in this study observed that decrease in chlorophyll content and increase in proline content. Authors should discusses the importance of these changes in plant defense with strong evidences from previous studies.
Answer: According to the proposal of Reviewer 2, the discussion part was modified and appropriately expanded.
“The chlorophyll content in plant tissues is a key factor in interactions between plants and insects [33]. Changes in chlorophyll concentrations occur in response to a wide range of stresses, including biotic stresses such as insect feeding and pathogenic infections [34]. Our results support the observations of Huang et al. [41], as we confirmed a relative decrease in chlorophyll content in Populus tremula in response to both sucking and chewing insect attacks. Photosynthesis is the main source of ROS in light, reactions in chloroplasts produce a variety of ROS forms, including singlet oxygen, superoxide, and hydrogen peroxide at high rates even under optimal conditions [30, 31]. ROS have deterring effect on the plant DNA and protein complexes. At the same time, insect herbivory has a negative impact on leaf water status, resulting in stomatal closure [61]. Therefore, it is possible that plants suffering from insect herbivory may also experience excess excitation energy caused by an excessive concentration of singlet oxygen [45]. Given the extensive cross-talk between light, ROS, and hormonal signalling, this phenomenon is likely to have a strong impact on plant responses to insect herbivores. In this scheme, increasing concentrations of certain phenolic compounds, known for their scavenging activity in eliminating ROS, appear to be advantageous [1].
“The work of Lackner et al. [49] provides insight into the dynamics of proline in Lymantria dispar, which, according to their observations, has a phagostimulatory effect which implies that leaves with higher levels of proline are preferred by spongy moths. According to our results, the activity of this herbivore does not induce increased levels of proline in Populus tremula. Proline is considered a reliable marker indicating plant stress due to drought [85, 87], reaching up to three times the concentration [21]. In this context, there is a potentially greater risk of spongy moths attack in poplar stands suffering from drought. As part of the aphid treatment, there was apparently a significant increase in proline content due to the consumption of phloem sap by aphids.”
Authors claimed that the some volatile contents decreased during the aphid infestation. This phenomenon has to be discussed in the context of plant defense against insects.
Answer: During the analysis of VOCs, within the statistical model, we focused on determining the separation framework within 3 treatments - reference, aphids infested, and moths infested. The order of compounds according to Table 2 and Table 3 for each treatment is related to concentration to highlight the importance of individual VOCs according to VIP plots for each treatment compared to the reference. The degree of separation of compounds according to the feeding guild of insects is, in our opinion, nicely displayed in Figures 2, 3, 4. Despite all efforts, we were unable to find sources in the literature that would explain the cause of the decrease/increase of specific VOCs for aphids/moths damaged leaves. Therefore, we discussed substances and their effects according to the highest VIP values. This question is thus a suitable and valuable topic for further research.
Materials and methods.
Did authors confirmed the “genetically homogenous” condition of the plants?
Answer: Yes, we confirm that the plants we worked with were genetically uniform. As described in the methodology and outlined in Appendix A, we conducted controlled crossbreeding and collected seeds, from which all individuals for our experiment were propagated using somatic embryogenesis techniques.
Please write the number and stage of the insects released on to the plants.
Answer: These data have been added to the methodology.
Why authors specifically used “Scheffe test” post-hoc test, as their sample number/size of the experiment is equal?
Answer: The size of the same number of n in each treatment was the reason for using the “Scheffe test".
Comments on the Quality of English Language
The manuscript requires English language editing to improve its quality. In many places the phrases are difficult to understand, they could be re written simple way for better understanding.
Answer: The English quality was again checked by a native speaker, the text was checked to improve clarity
Please see the attachment.
